# Macroscopic Evaluation of Colon Cancer Resection Specimens

**DOI:** 10.3390/cancers15164116

**Published:** 2023-08-15

**Authors:** Ross Jarrett, Nicholas P. West

**Affiliations:** Pathology & Data Analytics, Leeds Institute of Medical Research, St. James’s University Hospital, School of Medicine, University of Leeds, Leeds LS9 7TF, UK

**Keywords:** colon cancer, pathology, quality of surgery, feedback, macroscopic assessment

## Abstract

**Simple Summary:**

Colon cancer is a common disease that is primarily treated by surgically removing the affected bowel, but the quality of surgery is variable internationally, leading to suboptimal patient outcomes. Pathologists should provide feedback to surgeons and can help to improve long-term patient outcomes. This review summarises the key aspects of pathological quality control that should be adopted internationally to improve the chances of survival from this deadly disease.

**Abstract:**

Colon cancer is a common disease internationally. Outcomes have not improved to the same degree as in rectal cancer, where the focus on total mesorectal excision and pathological feedback has significantly contributed to improved survival and reduced local recurrence. Colon cancer surgery shows significant variation around the world, with differences in mesocolic integrity, height of the vascular ligation and length of the bowel resected. This leads to variation in well-recognised quality measures like lymph node yield. Pathologists are able to assess all of these variables and are ideally placed to provide feedback to surgeons and the wider multidisciplinary team to improve surgical quality over time. With a move towards complete mesocolic excision with central vascular ligation to remove the primary tumour and all mechanisms of spread within an intact package, pathological feedback will be central to improving outcomes for patients with operable colon cancer. This review focusses on the key quality measures and the evidence that underpins them.

## 1. Introduction

With global diagnoses of more than 1.8 million new cases per year, colorectal cancer is the third most frequent malignant disease seen in male and female populations [1]. When magnified further, the data show an uneven split between colon and rectal cancers of approximately 70% and 30%, respectively [2]. Colorectal cancer is a good candidate for surgical intervention with a curative aim [3], and the way in which the resection specimen is evaluated is fundamental within this context. Indeed, the evaluation by pathologists can facilitate changes to standardised surgical procedures, resulting in superior patient outcomes.

Initially described by Heald, total mesorectal excision (TME) is now the international standard approach for surgically resecting rectal cancers [4]. The paradigm shift that TME created has not only vastly improved oncological outcomes via a standardised surgical approach, but also, importantly, utilised pathological evaluation and feedback [5,6]. An identical principle for the resection of colon cancer is the recently described complete mesocolic excision (CME) with central vascular ligation (CVL) [7]. Pioneered by Hohenberger and colleagues in Erlangen, Germany, this approach has also led to significant improvements in oncological outcomes. These outcomes have been replicated in other centres utilising the technique [8,9,10,11]. Although CME with CVL provides many of the same oncological benefits of TME, it is still yet to be adopted as an international surgical standard, with many centres still favouring ‘conventional’ approaches largely due to concerns over morbidity with high-tie surgery [12,13,14].

## 2. Paving the Way: Jamieson and Dobson

As far back as 1909, well before CME was described, Jamieson and Dobson first described the lymphatic drainage of colon cancer and the optimal principles for surgical resection [15]. Their proposal relied on the fundamental principle that colon cancer lymphatic drainage follows the arterial supply. Thus, to prevent any remaining cancer from being left in situ and disease recuring locally or distally post-surgery, ‘whole-package’ resection is required, including the tumour and associated lymphatics, which brings benefits by reducing the risk of spread as well as preventing tumour spillage. This is exactly the same principle that TME is built upon [4,7].

Anatomically speaking, the mesocolon and mesorectum are continuous structures [16,17,18]. Similarly, the embryological tissue planes of these structures are analogous. Each is contained within a peritoneum and visceral fascia; ‘whole-package’ resection in both CME and TME is made possible due to the presence of these planes [4,7]. One important difference is the marked variability in the vasculature of the colon compared to the rectum [7,19]. This poses idiosyncratic surgical challenges, which three-dimensional computed tomography (CT) angiography has been suggested to overcome [20,21].

Whilst Jamieson and Dobson diligently charted the lymphatic drainage of colon cancer over one hundred years ago, more recently, the Japanese Society for Cancer of the Colon and Rectum (JSCCR) published a system to more thoroughly classify colon lymph node anatomy based on their proximity to the wall of the bowel [15,22]. Initially, a cancer will drain to the paracolic (D1) nodes, which are closest to the bowel wall and follow the path of the marginal arteries. Intermediate (D2) nodes represent the next drainage layer, and these follow the path of the branches of the superior and inferior mesenteric arteries. Finally, the highest level of drainage potentially included in standard specimens is provided by the central (D3) nodes. These nodes are found on both the right and left sides of the colon. On the right, they reside at the origin of the ileocolic, right colic and middle colic arteries, and on the left side, they run along the inferior mesenteric artery from the origin at the aorta to the branch of the left colic artery [22].

## 3. Contemporary CME with CVL

According to Hohenberger, there are three key components to CME with CVL. Sharp dissection should be achieved along the embryological tissue planes, whereby the visceral and parietal fascia are separated, resulting in an intact surgical specimen lined by peritoneum and fascia, as applicable. The primary aim is to ensure this lining remains intact, thus reducing the risk of tumour spillage. Next, to facilitate the removal of the D3 lymph nodes and maximise the lymph node harvest, CVL should be performed on the supplying colonic arteries. This may also be referred to as “high-tie” and involves ligation of the vessels close to their root. The level of venous ligation generally does not receive the same attention in the literature as the arterial tie, but Hohenberger confirms that the veins should also be ligated centrally. The scientific rationale for this is less clear. Hohenberger finally specifies that by resecting an adequate length of bowel, more longitudinal paracolic nodes will be removed [7,23]. A lack of randomised trials means the relative significance of each of these components is not yet fully understood.

In Japan, the JSCCR also recognises the importance of operating within the embryological plane, although they do not undertake CVL on all patients. The decision on the level of vascular ligation is made inter-operatively and largely depends on the depth of primary tumour invasion [22]. More specifically, they advise that T1 tumours undergo D2 resection with intermediate vascular ligation, while T2-4 undergo D3 resection with CVL [22,24].

What is deemed an adequate length of longitudinal bowel resection from the primary tumour is also a point of difference between Hohenberger and the JSCCR. The latter advocate a more conservative length (rarely more than 10 cm), while the former is more radical, going at least one vascular arcade beyond the tumour, which results in a longer length of bowel resected [24,25]. There are several Japanese studies that support a more conservative approach. Longitudinal lymphatic spread beyond the paracolic area in left- and right-sided tumours was found to be as low as 0% and 1–4%, respectively [26,27]. In contradiction to this, another study found that in 16% of cases, the first metastatic lymph node was found in the paracolic region, in excess of 5 cm from the primary tumour [28]. Despite this, West et al. found that while the length of resection was variable, there was a high rate of mesocolic plane excision in both Japanese and Hohenberger cases. The long-term outcomes for both were also analogous, leading to the suggestion that the operating plane is likely to be of greatest significance, with the height of the ligation having a smaller influence in some cases [24]. A comparison of CME, D3 and conventional surgery is shown in Figure 1.

A further development is the expanded use of laparoscopic and robotic-assisted techniques [29]. Concerning CME, several retrospective comparative studies and a randomised controlled trial have been conducted to compare open versus laparoscopic techniques [30,31,32,33,34,35,36]. Findings were consistent across these studies; operating times, oncological benefit, safety profile and specimen quality were similar between the two groups. For D3 resection, a Japanese randomised controlled trial was conducted to compare open versus laparoscopic techniques; oncological outcomes were equivalent, but lower complications and shorter hospitalisation were apparent [31,36]. Other retrospective comparative studies also concur with this [29,32,33,35]. While robotic-assisted CME has predominantly been utilised for right-sided procedures, studies conducted show that despite a laparoscopic or open environment, the oncological benefit and safety profile results are similar [37,38]. However, although robotic CME increased operating time, it also increased lymph node yield and was associated with low conversion rates [37,38,39,40]. Initial findings show encouraging results in this area.

## 4. Oncological Outcomes and Benefits

Hohenberger et al. reported a series of cases performed over 24 years, during which there were defined periods of analysis: pre-CME (1978–1984), development of CME (1985–1994) and implementation of CME (1995–2002). Between the pre-CME and implementation periods, 5-year cancer related survival after surgery increased by 6% (82.1% to 89.1%) and 5-year recurrence rates decreased by 2.9% (6.5% to 3.6%) [7]. Notwithstanding these significant results, there are some limitations with the series; the duration over which the study was conducted carries potential confounders, and as it was a non-randomised study, exactly how patients were selected is unclear. However, significantly increased disease-free or disease-specific survival as a result of CME has been shown in several subsequent studies [8,9]. Decreased rates of local and distal recurrence have also been shown [10,11]. However, this is not a universal picture, as other studies have shown no significant difference in overall survival with the use of CME; however, pathological quality control is not uniform [9,41,42,43]. In addition, increased morbidity has been associated with CME use, especially in right-sided tumours [22,24], although this has been disputed by a recent meta-analysis [44].

To date, there have been only retrospective studies published that compare CME with non-CME and, importantly, no randomised controlled trials. With evidence increasingly suggesting that CME is largely effective due to an integral mesocolic plane, it is essential that independent pathological analysis take place on resected specimens. Indeed, it is notable that many studies conducted in this area do not consider this an essential component. However, two important randomised controlled trials are ongoing. COLD compares D2 and D3 lymph node dissection in colon cancer [45], and RELARC compares D2 dissection with CME for laparoscopic right hemicolectomy in colon cancer [46].

Some studies looking at CME have reported results that encourage its use. Two studies reported that surgeons employing the technique regularly are more likely to produce a high-quality specimen compared to those that do not [25,47]. Another study found that 88.5% of specimens were judged to be in the mesocolic plane following CME, whereas conventional surgery was found to be 47.4% [25]. In addition, Ng et al. found that at a large referral centre, two-thirds of all colon cancer resections were performed in the mesocolic plane [14]. It is therefore all the more pertinent that West et al. found a 15% improvement in 5-year overall survival for patients with resections in the mesocolic plane versus those in the muscularis propria [48]. This benefit increased to 27% for stage III disease. A large review considering 18,989 patients from 27 studies also found significant positive impacts on 3-year and 5-year overall survival as well as 3-year disease-free survival [49].

One can argue that simple standardisation of a surgical procedure provides further rationale to adopt CME as the convention; studies have elicited better quality resections via standardisation [50,51], which can facilitate better patient outcomes [52,53]. Further significant results from a Danish study were found in which one hospital underwent a CME education programme and was compared to five others that did not; in the CME educated group versus the conventional group, mesocolic plane excision was 75% vs. 48%, distance between tumour and ligation point was 105 mm vs. 84 mm, mean lymph node yield was 28 vs. 18 and improved 4-year disease-free survival was 85.8% [95% CI 81.4–90.1] vs. 73.4% [66.2–80.6] [9,50]. When the original programme was rolled out across the other five sites, mesocolic plane excision increased from 58% to 70% (*p* < 0.001) [54]. It is also apparent that through such education programmes, CME can be relatively quickly learned and adopted [50,55].

## 5. The Pathologist, a Gatekeeper to Quality Control

The pathologist is uniquely suited to the role of assessing the quality of colon cancer resection specimens. Not only are they independent of the surgery, but they are also exposed to specimens from multiple surgeons. When harnessed by the MDT, this essential experience facilitates a rich feedback loop, wherein the pathologist is able to supply direct feedback to the MDT, including photographic records. The role has already been exemplified in TME and is now considered standard of care following rectal cancer surgery. The Medical Research Council CR07 trial demonstrated an improvement in the quality of resected specimens over time when assessing how the plane of surgery affected local recurrence in rectal cancer. Between 1998 and 2005, mesorectal plane excisions increased from less than 50% to more than 60%, muscularis propria excision decreased from over 20% to 10%, and circumferential resection margin involvement decreased from 21% to 10% [6]. Pathological quality control and direct feedback to the MDT during the trial are likely to have contributed to improving the quality of the resected surgical specimen over time. Although other factors, such as the introduction of MRI for surgical planning and adjuvant chemotherapy, have played their part in the sustained improvements in long-term rectal cancer survival, the improvement in specimen quality should not be understated [30].

## 6. Pathological Assessment of Mesocolic Integrity

In CME, the integrity of the resected mesocolon should not be breached. This is described as surgery in the mesocolic plane. This results in the cancer and potential mechanisms of spread (tumour deposits, lymphatics, lymph nodes, nerves and blood vessels), being contained within a package lined by the peritoneum and fascia [7,15]. The requirement to be scrupulous concerning mesocolic integrity is founded in studies that show improved patient outcomes resulting from CME [7,36]. The integral package brings significant benefits by reducing the risk of intra-abdominal recurrence in two ways. Firstly, it prevents spillage of tumour cells into the peritoneal cavity, and secondly, it increases surgical radicality [48]. In a single-centre retrospective study, mesocolic versus muscularis propria plane surgery conferred a 15% overall survival advantage at 5 years for mesocolic plane patients [48]. In addition, the advantage increased to 27% in stage III patients, showing that it is even more crucial to remove the mesocolon intact at the later stages of disease.

Largely based on the CR07 trial [5,56] and subsequently developed for the MRC CLASICC trial [57], a three-tier system is used to grade the plane of surgery (see Table 1 and Figure 2). This provides an objective specimen quality assessment and facilitates the evaluation of patient prognosis. The assessment should first be performed on the intact, fresh specimen. Then, a secondary assessment should commence on the intact and cross-sectionally sliced formalin-fixed specimen to establish the presence of any defects as well as their respective depths. The optimal plane of surgery is represented by the mesocolic plane, whereby the peritoneal and fascial linings are intact. Only defects that are less than 5 mm in depth are permitted at this grade. Intermediate quality is represented by the intramesocolic plane. At this grade, the depth of any single defect must be in excess of 5 mm but must not extend down to the muscularis propria. A poor-quality specimen is represented by the muscularis propria plane, in which a defect is found that extends on to the muscularis propria or deeper, e.g., surgical perforation. Importantly, a final grading is governed by considering the poorest quality area, even if it is limited in size [48].

The process of mesocolic grading is prone to a degree of intra- and inter-observer variation. Munkedal et al. found that inter-observer agreement was poor (k < 0.4), while intra-observer agreement was fair to good (k 0.4–0.7) [58]. They proposed several refinements to the grading system to improve reproducibility, one of which drew on the Japanese classification of lymphatic drainage: only the mesocolon in the tumour lymphatic drainage field should be assessed. However, they suggest excluding from evaluation the area within 10 mm of the longitudinal margins, regardless of inclusion in the drainage area. This is due to consistent irregularity at the margins, which risks unreliability. Finally, on some specimens there are ‘peritoneal windows’ (fused serosal layers devoid of intervening fat), which if damaged in isolation should also not be a cause for downgrading [58]. Identifying and minimising inter-observer variation is essential to avoiding bias; thus, it is paramount to centrally moderate specimen grading within a clinical trial setting. The UK FOxTROT trial is an exemplar of this; locally across 80 centres, pathologists graded specimens, and these will be compared centrally to facilitate calculation of inter-observer variation [59].

## 7. Distance between the Tumour and Point of Central Arterial Ligation

A quality measure of CVL is the distance between the tumour and the arterial ligation point on the pathological specimen. This measure is flawed at the individual level, as the length of the original vessel is not known by the pathologist, and the vessel will contract after removal and even more with formalin fixation [47]. However, from the population perspective, when intermediate-level ligation was compared to central ligation by Hohenberger, the latter was associated with a significantly greater distance between the tumour and ligation point: 81.4 mm vs. 128.7 mm in right-sided tumours; *p* < 0.0001 and 97.0 mm vs. 145.0 mm in left-sided tumours; *p* < 0.001. It was also significantly correlated with a greater lymph node yield [47]. Further studies have corroborated these findings [11,25,50], and associations have been made between optimal-plane specimens and an increased distance between the tumour and high-tie, as well as increased lymph node yield [11,60].

The central ligation height is a key marker for the radicality of CME, but there are challenges. The measure is best conducted on fresh specimens immediately after resection. However, for logistical and clinical reasons, this is often impractical [61]. To optimise this measurement, two solutions have been suggested: in-theatre measuring of the specimen or photography against a metric scale. In addition, there is significant anatomical variation found in the length of the central vessel [25,61], and thus, at the individual level, its measure holds little value. It is, however, useful at the population level because it accurately conveys the radicality of CME and can therefore be used for audit and training.

Assessment of the remaining arterial stump radiologically is another consideration. Despite Swedish and U.K. studies, which cite anatomical variation as reason to suggest the length of the remaining arterial stump does not accurately predict the length of the resected vessel [62,63], stump length is still a strong marker for surgical radicality. A Danish study quantified that the mean stump length measured by CT was 38 mm (95% CI: 33–43 mm), whereas the target within the CME with CVL context is approximately 10 mm [61]. If increased radicality improves outcomes, there is potential to use this post-operatively for immediate feedback, as well as for follow-up, because the arterial stump does not significantly change length over time [64]. However, a CT conducted immediately after surgery is unlikely to be palatable due to increased radiation exposure, so this may need to be done on the routine 12-month post-operative scan.

## 8. Length of Bowel Resection

A multi-centre study in Japan (where the ‘10 cm rule’ is standard) compared various surgical approaches (CME with CVL, D3 lymph node removal and conventional surgery) in stage III colon cancer and considered bowel resection length [25]. The study found CME bowel length was significantly greater than D3 (median 355 mm vs. 184 mm in right-sided tumours; *p* = 0.0003, 355 mm vs. 146 mm in left-sided tumours; *p* < 0.0001), but the central arterial ligation height was not significantly different (median 115 mm vs. 103 mm in right-sided tumours, 128 mm vs. 120 mm in left-sided tumours). When conventional ‘intermediate ligation’ surgery is compared to D3, the latter yields significantly shorter bowel resection length and greater arterial ligation height (median central ligation height: 81 mm vs. 103 mm in right-sided tumours; *p* = 0.037, 100 mm vs. 120 mm in left-sided tumours; *p* = 0.034). Unsurprisingly, surgery performed in the conventional way was in the mesocolic plane only 47% of the time versus 72% for D3 surgery [25]. This suggests that, when compared to other CME parameters, the oncological value gained by increasing the length of resection is relatively limited.

Kobayashi et al. found that as the length of bowel resection increases, so does the number of resected lymph nodes [25]. Although this has been corroborated in other studies [24,25,47,48], the number of malignant lymph nodes does not appear to increase, suggesting that there is no benefit to removing large numbers of benign nodes that are well away from the tumour site [24,25]. Longitudinal outcomes from Hohenberger and those in Japan both show comparable results despite significant differences in specimen length. The T-REX study aims to establish a more accurate comparison of optimal resection length and ligation height in centres around the world undertaking both CME and D3 surgery [65].

## 9. Lymph Node Yield

In general, CME lymph node yield is greater than that of non-CME surgery [66]. Two studies by West et al. quantified CME vs. non-CME: a median of 18 versus 30 nodes [47] and a median of 18 versus 28 nodes [50]. Due to the longer bowel resection in CME, many of the additional nodes are more likely to be of the D1 and D2 types, although the D3 nodes will also be included in CME [47]. There has been international debate as to the optimum number of nodes to harvest and how they should be processed in an effort to stage colorectal cancer accurately. The Royal College of Pathologists advises that every node within the specimen should be examined and that pathologists should regularly audit their practice with the expectation of a minimum average of 12 nodes across 50 cases [67]. However, leading centres regularly harvest 20–40 nodes via careful dissection or the use of ancillary methods such as methylene blue [68]. CME is thought to partially bring benefit from maximised node yield via increased accuracy of staging. Hohenberger found that a yield of 28 was independently associated with 5-year cancer-related survival (96.3% vs. 90.7%, *p* = 0.018) for those patients that were node-negative [7]. Another study found CME to have a significant benefit over non-CME surgery in stage I and II colon cancer for similar reasons [9]. Part of the survival benefit from greater lymph node numbers in early-stage disease arises from stage migration. Much of the benefit is believed to be due to a better immune response, e.g., the relationship between deficient mismatch repair/microsatellite instability, immune response and patient outcomes [69].

A primary reason why staging is of utmost importance is for patient access to adjuvant chemotherapy; low yields reduce the likelihood of a stage III diagnosis, which then precludes access to such therapy [70]. However, node yield as a marker of surgical quality in isolation is not recommended unless the surgical plane is also considered, as even the most radical surgery can occasionally result in low yield, and conversely, specimens that are extensively disrupted may still contain many nodes [71]. Most important is that an exhaustive search be performed that evaluates all paracolic, intermediate and central nodes within 10 cm of the tumour. Morris et al. also investigated whether the practicing pathologist had any influence over the yield; it was found that specialist pathologists (those that regularly undertake this work) were more likely than non-specialists to achieve adequate yield within their specific role (OR, 2.16; 95% CI, 1.93 to 2.41) [70].

## 10. Photography

The key to recording and documenting specimen quality is standardised specimen photography. This can be utilised by the MDT to facilitate audits of surgical quality, scientific discussion and training. The aforementioned feedback loop is strengthened by this activity; studies show that benefits have been gained in the assessment of specimen quality as well as linked to improved outcomes, especially when used in clinical trial settings [34,72]. More specifically, Munkedal et al. found that by utilising specimen photography within MDT meetings wherein surgeons could derive feedback, mesocolic plane surgery increased (52% to 76%, *p* = 0.02) [34]. Photography also allows the addition of centralised mesocolic grading across multiple centres, e.g., in a clinical trial, which in turn has been shown to reduce interobserver error [24,25,50]. The recommended standardised approach is outlined below (see Figure 3).

Photography should be high-resolution and taken directly above the specimen to reduce distortion artefacts. A fixed stand should ideally be used to mitigate against movement artefacts. Placed next to the specimen should be a metric scale, as this allows for calibration of the software for morphometrical analysis. Ideally, prior to formalin fixing or opening, the anterior and posterior aspects of the whole specimen should be photographed. The mesentery should be laid out flat and without tension, so that the proximal and distal aspects, tumour and vascular ties can be labelled or obviously visualised. Close-ups of any relevant areas should be taken, e.g., mesocolic defects and perforations. Formalin fixation should follow, and the tumour segment should be sliced at 3–4 mm intervals. The slices should be laid out sequentially, with the proximal and distal slices labelled. Finally, clinically important areas should be noted, and close-up photographs taken. This protocol has been used to quality control specimens for a number of studies referenced in this manuscript [24,25,34,47,48,50].

## 11. Conclusions

TME has improved oncological outcomes and increased overall survival for rectal cancer patients around the world. This has predominantly been achieved by operating within the mesorectal plane, which facilitates intact resection of the cancer. Unsurprisingly, therefore, TME is now internationally standardised and precisely defined. By applying the same principles, CME can provide similar benefits for the resection of colon cancer. Studies to pathologically assess the quality of resection have confirmed that when CME is compared to conventional surgical approaches, the former produces a higher-quality specimen (more likely to be intact and within the mesocolic plane) and facilitates improvements in other quality markers. These include increased lymph node yield and improved radicality of surgery by centrally ligating the supplying vasculature. Operating within the mesocolic plane has been associated with an overall improvement in 5-year survival, especially for patients with stage III colon cancer.

Notwithstanding the suitability of the pathologist for objectively, systematically and promptly reporting an independent assessment of the specimen, it is unfortunate that recent studies comparing CME with non-CME have negated the importance of such an analysis. Lack of standardisation in CME means that inclusion of this analysis in the future is even more important and should be mandatory to interpret trial data. These benefits are only increased further when combined with a well-functioning MDT. Not only can pathological data inform MDT discussions around the individual patient, but it can also be extensively used in training sessions, informing best practices and continually improving the MDT as well as overall patient outcomes. It is therefore essential that any new clinical trials in this area be underpinned by independent pathological quality control.

## Figures and Tables

**Figure 1 cancers-15-04116-f001:**
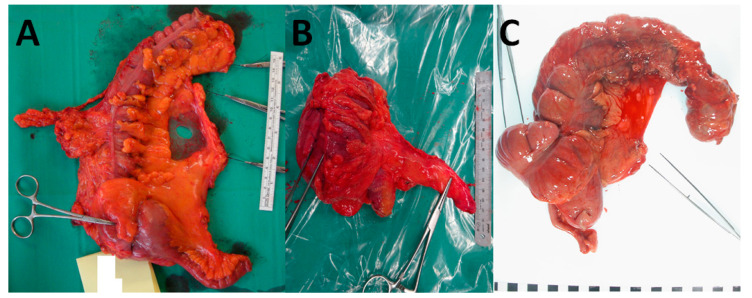
A comparison of CME, D3 and conventional surgery, showing a right hemicolectomy resected through CME with CVL (**A**), D3 resection (**B**) and conventional D2 resection (**C**). Note the increased length of the vascular pedicle with both CME and D3 surgery compared to conventional D2 surgery. Also note the significantly longer length of the colon removed with the CME approach compared to D3.

**Figure 2 cancers-15-04116-f002:**
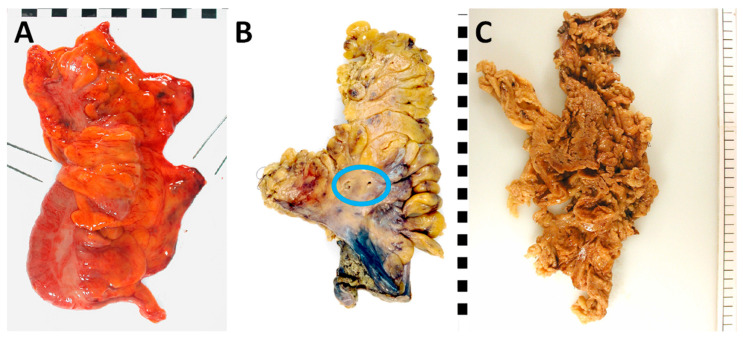
Pathological assessment of the mesocolic plane, showing examples of specimens in the mesocolic plane (**A**), intramesocolic plane (**B**) and muscularis propria plane (**C**). Not the small mesenteric disruptions in B that do not extend down to the muscularis propria (blue circle). The specimen in C shows a ragged mesentery with multiple disruptions down to the muscularis propria.

**Figure 3 cancers-15-04116-f003:**
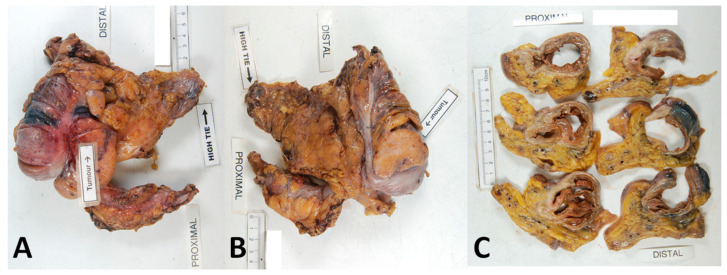
Optimal specimen photography protocol. Three separate images should be captured as a minimum. These include the anterior whole specimen view (**A**), posterior whole specimen view (**B**) and cross-sectional slices (**C**).

**Table 1 cancers-15-04116-t001:** Pathological assessment of the mesocolic plane.

Plane	Description
Mesocolic plane	Mesocolon intact and covered by peritoneum and fascia (where relevant). Defects measure no more than 5 mm in maximum size.
Intramesocolic plane	Mesocolic defects greater than 5 mm in size that do not extend down to the muscularis propria.
Muscularis propria plane	Substantial mesocolic defects that extend down on to the muscularis propria or beyond, e.g., perforation.

## Data Availability

Not applicable.

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
