# Peer review of "Macroscopic Evaluation of Colon Cancer Resection Specimens"

_cancers, 2023, doi:10.3390/cancers15164116_

Round 1
Reviewer 1 Report
Thank you for the opportunity to review your work.
Honestly, I was in need of finding a work with such an eloquent and objective review of what CME is today. I really enjoyed reading your work.
I make a few notes:
1 - I think the work could be more complete with the use of some figures, namely, of a surgical specimen photographed correctly, of a specimen fixed with formalin with the slices made correctly, photos of different qualities of mesocolon. Many of the works cited have these photographs available, in any case I don't think you will have any difficulty in finding them in everyday practice.
2 - As mentioned, we always talk about lymphatic drainage in relation to the arteries. However, in the definition of D3, and for some, CME, there is a general notion that the surgeon should skeletonise the vein and go to the origin of the venous structures. It would be interesting to discuss this in the paper justifying why this is the case.
3 - It would be interesting to see in the text what you consider to be the reason why bowel lenght is lower in D3 than in CME in the study with reference 25.
4 - Can you comment on why the increase in the number of nodules removed is only associated with better survival in earlier stage patients, as per the study presented (7)?
5 - In the second paragraph of section 9. Photography, the authors describe how the treatment of the surgical specimen should be carried out, please cite the reference of the method in question.
Author Response
Response: We thank the reviewer for their kind comments.
- We agree that some figures would help to reinforce the key messages regarding the photography protocol and specimen quality. We have now added three figures. Figure 1 shows the differences between CME, D3 and conventional D2 surgery. Figure 2 shows examples of mesocolic, intramesocolic and muscularis propria plane surgery. Figure 3 demonstrates the optimal photography protocol.
- We agree that the level of venous ligation does not receive the same attention as the level of arterial ligation in the literature. Hohenbergers original paper certainly states that the veins should also be ligated centrally, although the scientific rationale for this is less clear. A new comment has been added to the manuscript in section 3 on page 2.
- There is a clear difference in bowel length between the Japanese D3 and German CML with CVL specimens. This stems from a difference in concept. With Japanese D3 they aim for 10cm length of bowel beyond the tumour. With Hohenbergers description of CME with CVL, they aim to take the next vascular arcade beyond the tumour, which results in a longer length of bowel resected. This is already stated in section 3 at the bottom of page 3, but a short addition has been made for clarity.
- It is interesting that there is a relationship between the number of lymph nodes resected and survival, even in node negative disease. More lymph nodes examined results in more accurate staging, so a proportion of this effect will be due to stage shift. But we believe the greatest effect is due to biology with patients having a higher lymph node yield a better immune response and therefore survival. It is recognised that the immune response in deficient mismatch repair patients is greater and associated with larger and greater number of lymph nodes. dMMR is also well recognised as a good prognostic feature. A comment has been added to section 9 on page 7 with a new reference.
- A number of studies referenced in the manuscript are based on this specimen photography protocol. This has been clarified in the text.
Reviewer 2 Report
This review provides a satisfactory examination of Macroscopic Evaluation of Colon Cancer Resection Specimens, covering the surgical procedure to pathology macroscopic evaluation, which is a standard practice. However, it must be noted that the topic does not fall into the realm of high priority. Nevertheless, certain aspects, such as the measurement of the distance of CVL from the tumor, hold practical significance. On the whole, the content is valuable for publication. A minor comment is presented below:
It is preferable to include a few figures that emphasize the surgical procedure and the resected gross pathologic specimens, focusing on the key points that require attention
Author Response
Response: We thank the reviewer for their comments. Whist some of the elements of colon cancer dissection are considered standard practice, the manuscript describes evidence based best practice that is not performed internationally to date. Rectal cancer surgery and pathology are relatively well standardised, but this is not the case for colon cancer and explains why in several countries, colon cancer outcomes are now worse than that for rectal cancer. Mesocolic grading, assessing central radicality and specimen photography are not routinely applied, even in the UK. This article promotes their importance, which we strongly believe will help surgeons to improve the quality of specimens and therefore improve patient outcomes.
We agree that some figures would help to reinforce the key messages. We have now added three figures. Figure 1 shows the differences between CME, D3 and conventional D2 surgery. Figure 2 shows examples of mesocolic, intramesocolic and muscularis propria plane surgery. Figure 3 demonstrates the optimal photography protocol.
Round 2
Reviewer 1 Report
Thank you for the clarification.
Do the authors own the figures (figure 1) if not they should have references. Anyway, they should compare surgeries for the same tumor location. Are these cecal tumors? It doesn't look like figure B is a proper specimen of a D3 lymphadenectomy in a right hemicolectomy, perhaps a typhlectomy was performed.
Author Response
We thank the reviewer for their comments. All of the photographs used in the figures are from research studies led by the senior author. They have not been previously published. Figure 1 includes 3 images, which are all of caecal tumours resected by right hemicolectomy. Figure 1B is a specimen resected with the D3 approach in a Japanese hospital according to the JSCCR guidelines.